# Mortality in individuals treated with COVID-19 convalescent plasma varies with the geographic provenance of donors

Katie L. Kunze [1,15], Patrick W. Johnson [2,15], Noud van Helmond [3], Jonathon W. Senefeld [4], Molly M. Petersen [1], Stephen A. Klassen[4], Chad C. Wiggins [4], Allan M. Klompas [4], Katelyn A. Bruno [5], John R. Mills [6], Elitza S. Theel [6], Matthew R. Buras [1], Michael A. Golafshar [1], Matthew A. Sexton [4], Juan C. Diaz Soto [4], Sarah E. Baker [4], John R. A. Shepherd[4], Nicole C. Verdun[7], Peter Marks [7], Nigel S. Paneth[8,9], DeLisa Fairweather [5], R. Scott Wright [10,11], Camille M. van Buskirk[6], Jeffrey L. Winters [6], James R. Stubbs [6], Katherine A. Senese [4], Michaela C. Pletsch [4], Zachary A. Buchholtz [4], Robert F. Rea[10], Vitaly Herasevich [4], Emily R. Whelan [5], Andrew J. Clayburn[4], Kathryn F. Larson [10], Juan G. Ripoll [4], Kylie J. Andersen [4], Elizabeth R. Lesser[2], Matthew N. P. Vogt, Joshua J. Dennis[4], Riley J. Regimbal[4], Philippe R. Bauer [12], Janis E. Blair [13], Arturo Casadevall [14,16], Rickey E. Carter [2,16] & Michael J. Joyner [4,16✉]

Successful therapeutics and vaccines for coronavirus disease 2019 (COVID-19) have harnessed the immune response to severe acute respiratory syndrome coronavirus 2 (SARS-CoV-2). Evidence that SARS-CoV-2 exists as locally evolving variants suggests that immunological differences may impact the effectiveness of antibody-based treatments such as convalescent plasma and vaccines. Considering that near-sourced convalescent plasma likely reflects the antigenic composition of local viral strains, we hypothesize that convalescent plasma has a higher efficacy, as defined by death within 30 days of transfusion, when the convalescent plasma donor and treated patient were in close geographic proximity. Results of a series of modeling techniques applied to approximately 28,000 patients from the Expanded Access to Convalescent Plasma program (ClinicalTrials.gov number: NCT04338360) support this hypothesis. This work has implications for the interpretation of clinical studies, the ability to develop effective COVID-19 treatments, and, potentially, for the effectiveness of COVID-19 vaccines as additional locally-evolving variants continue to emerge.

A full list of author affiliations appears at the end of the paper.

Potential treatments to prevent coronavirus disease 2019 (COVID-19) and to ameliorate its disease course have converged on harnessing the immune response to severe acute respiratory syndrome coronavirus 2 (SARS-CoV-2). Despite the successful development of COVID-19 vaccines[1–3] and identification of COVID-19 therapeutics [e.g., convalescent plasma, remdesivir, monoclonal antibodies (mAbs), and steroids], there was an unexpected rise in global COVID-19 cases in late 2020 partially attributed to the emergence of several new SARS-CoV-2 variants that were specific to geographic regions[4,5]. Recent evidence suggests that SARS-CoV-2 exists as a variant distribution that evolves locally[6–8]. These small structural variations in SARS-CoV-2, which occur locally, may translate into immunological differences impacting the effectiveness of available treatments, and in some cases, COVID-19 vaccines have already demonstrated regionally varied effectiveness. For example, the chimpanzee adenovirus-vectored vaccine (ChAdOx1 nCoV-19) demonstrated 74% efficacy in the UK[9] but only 22% efficacy in South Africa[10]. The emergence of SARS-CoV-2 variants is a cause for concern, and vaccine and therapeutic strategies must account for local differences in transmissible SARS-CoV-2 variants.

Regional variants of SARS-CoV-2 were reported in the United States as early as November 2020 and may have been present earlier[11]. Early research has shown that local variants may impact the effectiveness of convalescent plasma, such that antibody responses to earlier viral strains are less effective against newer SARS-CoV-2 variants[12]. One of the perplexing findings observed with the use of convalescent plasma for COVID-19 is that observational studies have generally yielded favorable results, whereas randomized controlled trials have been less encouraging[13]. Large controlled clinical trials are more likely to use a central source of convalescent plasma, whereas observational studies tend to depend on a distributed network of blood collection facilities. The existence of differences in efficacy related to donor location could help to explain the wide variety of results observed in convalescent plasma studies.

Given that near-sourced convalescent plasma is likely to reflect the antigenic composition of local viral strains, we hypothesized that convalescent plasma has a higher efficacy when the donor and treated patient are in close geographic proximity. We evaluated this hypothesis in a US registry of 94,287 hospitalized COVID-19 patients who were treated with convalescent plasma from 313 participating blood collection centers. This allowed sufficient variability in donor–patient distance to test whether near-sourced convalescent plasma provides a survival benefit compared to distantly sourced convalescent plasma in transfused COVID-19 patients.

## Results
Of the 94,287 patients receiving transfusions through the Expanded Access Program (EAP) for convalescent plasma to treat COVID-19, 27,952 met inclusion criteria for this analysis (Fig. S1). Primary demographic and baseline characteristics of COVID-19 patients are reported in Table S1 stratified by geographic proximity of the plasma donation used to treat the COVID-19 patients [near- sourced convalescent plasma (≤150 miles) vs. distantly sourced convalescent plasma (>150 miles)]. Baseline characteristics were similar across distance cohorts except for geographic region, month of transfusion, race, respiratory failure, and low blood oxygen saturation as evidenced by standardized differences being ≥0.10. Treatment with azithromycin or steroids also had standardized difference ≥0.10, but these variables were not reported for a majority of the cohort. Figure 1 depicts the movement of convalescent plasma donations within and between US Census geographic areas[14] with both divisions and regions represented.

The rate of death within 30 days of transfusion for the entire cohort was 9.76% [2728 of 27,952; 95% confidence interval (CI), 9.42–10.11%]. Death within 30 days was lower in the group receiving near-sourced plasma [8.60% (1125 of 13,088; 95% CI 8.13–9.09%)] than in the group receiving distantly sourced plasma [10.78% (1603 out of 14,864; 95% CI 10.30–11.29%); (P < 0.001)]. Additional crude mortality rates delineated by patient characteristics, treatment and donation regions, and treatment month are included in Table S2 and are stratified by donor proximity. Notably, mortality rates within the near-sourced group were numerically lower in all cases, and in most cases, the 95% CIs did not overlap. The variable importance plot from gradient-boosting machine (GBM) analysis showed that the number of miles between the convalescent plasma collection and treatment facilities was the fourth most important predictor of death within 30 days of convalescent plasma transfusion (Fig. S2a). The partial dependence plot for distance displays the predicted probabilities of death within 30 days across the observed distances of plasma transport (Fig. S2b). Note that extreme distances were transformed in the GBM to set the upper limit at 2500 miles (i.e., winsorized).

Patients in the group receiving near-sourced convalescent plasma had a lower relative risk of death within 30 days of transfusion than patients receiving distantly sourced convalescent plasma (relative risk, 0.80; 95% CI 0.74–0.86). Adjusted regression models showed similar results (Table S3). Results of the additional analysis using a stratified data analytic approach further supported these findings by controlling for disease severity of the patient receiving convalescent plasma, time to convalescent plasma treatment from COVID-19 diagnosis or symptom onset, and convalescent plasma donor region (Fig. 2). These subgroupings capture the combination of U.S. Census region and the combined variable of time to treatment and disease severity (early administration with no complications, early administration with some complications, and late administration or with many complications). The pooled relative risk of death within 30 days of transfusion across the subgroups for near-sourced vs. distantly sourced convalescent plasma was 0.73 (95% CI 0.67–0.80).

## Discussion
In a large sample of COVID-19 patients aged 18–65 years transfused with convalescent plasma under the EAP, patients receiving near-sourced plasma exhibited lower mortality compared to patients transfused with distantly sourced plasma (8.6 vs. 10.8%). This trend was consistent across all regions of the US and persisted when controlling for other variables (e.g., patient characteristics, disease severity, and treatment methods). We interpret these observations to suggest that convalescent plasma donated from nearby COVID-19 survivors contained antibodies specific to local variants enabling greater viral neutralization and reduced mortality.

Our results are consistent with the biology of SARS-CoV-2 and the immunology of COVID-19. SARS-CoV-2 is an RNA virus that generates new variants through error-prone replication of its genome and thus exists as a constantly changing local variant distribution[9]. Over the past year, error-prone replication has led to the emergence of numerous major SARS-CoV-2 variants, some of which are much less susceptible to neutralization by antibodies elicited by earlier circulating strains[9]. These SARS-CoV-2 variants tend to attract attention when they replace the prior prevalent viral strains through increased transmission, mortality, and/or when they defeat vaccine immunity and antibody-based therapies through antigenic changes[15,16]. However, these major known variants are the proverbial "tip of the iceberg" for the genomic and antigenic diversity that exists for SARS-CoV-2. For example,

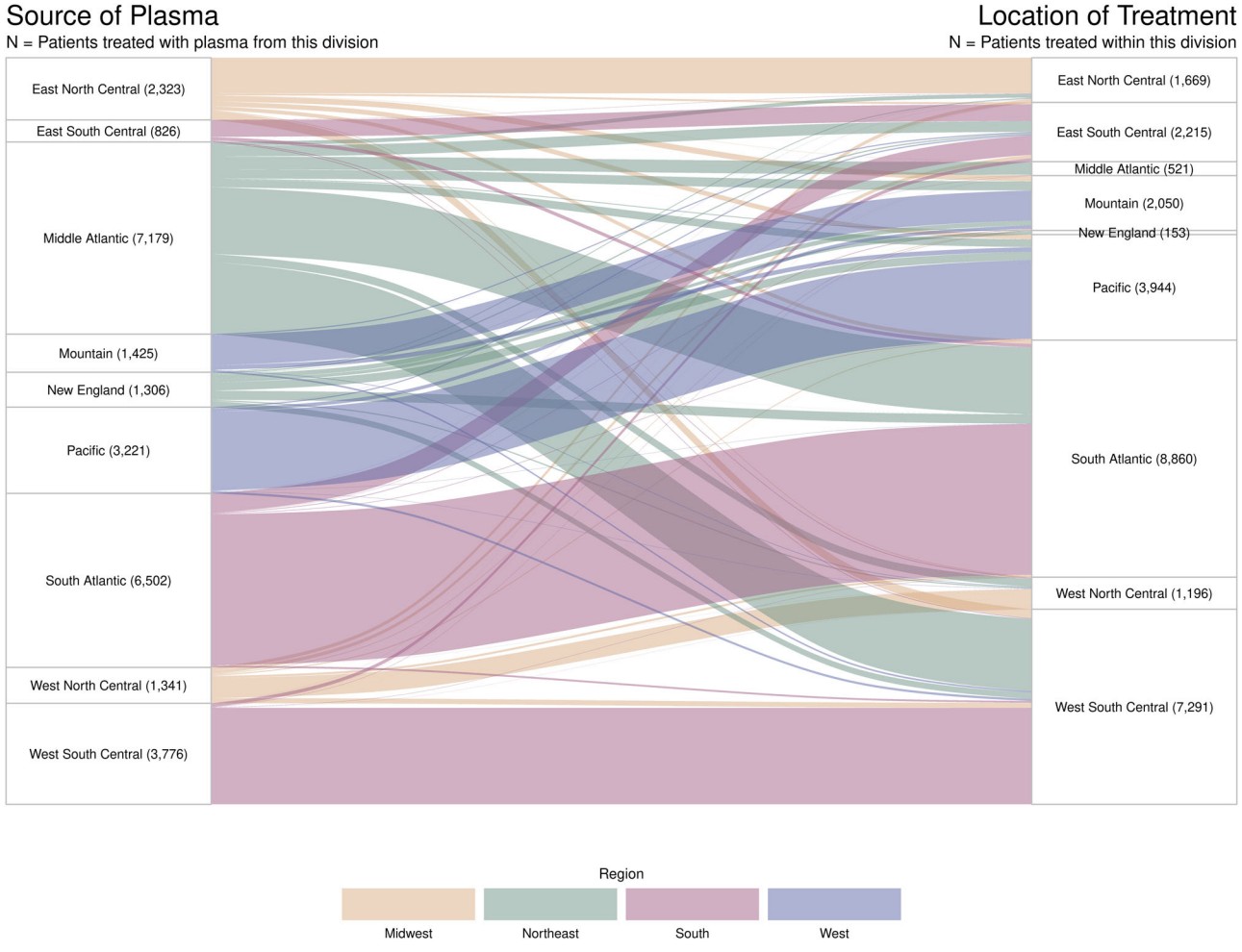

**Fig. 1 Sankey diagram of movement of convalescent plasma units between US Census regions and divisions.** Flow of convalescent plasma units from the location of their collection to the location of treatment is depicted by lines connecting divisions of the US. The width of each line is proportional to the number of patients treated. The color of each line represents the US Census region from which the convalescent plasma was donated. Note that low rates of transfusions in the Middle Atlantic and New England divisions, which make up the Eastern region, are due to a combination of the analysis time window and the exclusion of mechanically ventilated patients.

even within the Washington DC USA capital region different cities have different proportions of SARS-CoV-2 clades, implying tremendous regional diversity[17]. Hence, different communities can be expected to harbor distinct distributions of local variants of SARS-CoV-2 that, while insufficient to come to medical attention by virtue of not having acquired obvious new properties, in aggregate they could elicit different antibody responses that translate into convalescent plasma with varying antiviral capacity. In addition, these SARS-CoV-2 variants could have differential mortality rates, which may explain the high importance of region observed in the GBM. Stresses on local healthcare infrastructure and quality and availability of care during the waves of infection would also impact regional mortality rates. Regions of treatment and donor proximity are likely coupled due to convalescent plasma availability and distribution.

Our finding that near-sourced convalescent plasma was associated with lower mortality than distantly sourced convalescent plasma implies that small differences in the human immune response to local variants can translate to a major effect on therapeutic outcome. This can be particularly important for convalescent plasma where the active agent consists of polyclonal antibodies representing a complex mix of immunoglobulins that bind to many epitopes in viral proteins. This observation has far-reaching consequences. First, it provides indirect immunological

evidence for the notion that medically important variants were present in many US communities as early as the spring and summer of 2020. Second, it implies the superiority of locally sourced convalescent plasma for the therapy of COVID-19. Third, it suggests a biological explanation for the differences in efficacy between clinical studies that used locally sourced convalescent plasma vs. those that relied on central repositories. Fourth, as convalescent plasma continues to be used, these observations support a need to divert locally produced convalescent plasma for local needs and to increase collection in geographic areas with poor or no convalescent plasma collection capacity. Fifth, it implies that ensuring maximal efficacy from convalescent plasma will require better matching of antibody specificity to the local SARS-CoV-2 strain, which would require a more detailed characterization than simply measuring total and neutralizing antibody titer.

Our finding that the efficacy of COVID-19 convalescent plasma varies with the proximity of the donation site is novel and suggests that local differences in viral strain alter antibody neutralization capability. For both COVID-19 convalescent plasma and mAbs, the active ingredient is antibodies specific to SARS-CoV-2, yet these preparations vary in composition. mAb preparations are composed of one or two immunoglobulins that target defined epitopes on the SARS-CoV-2 spike protein, which

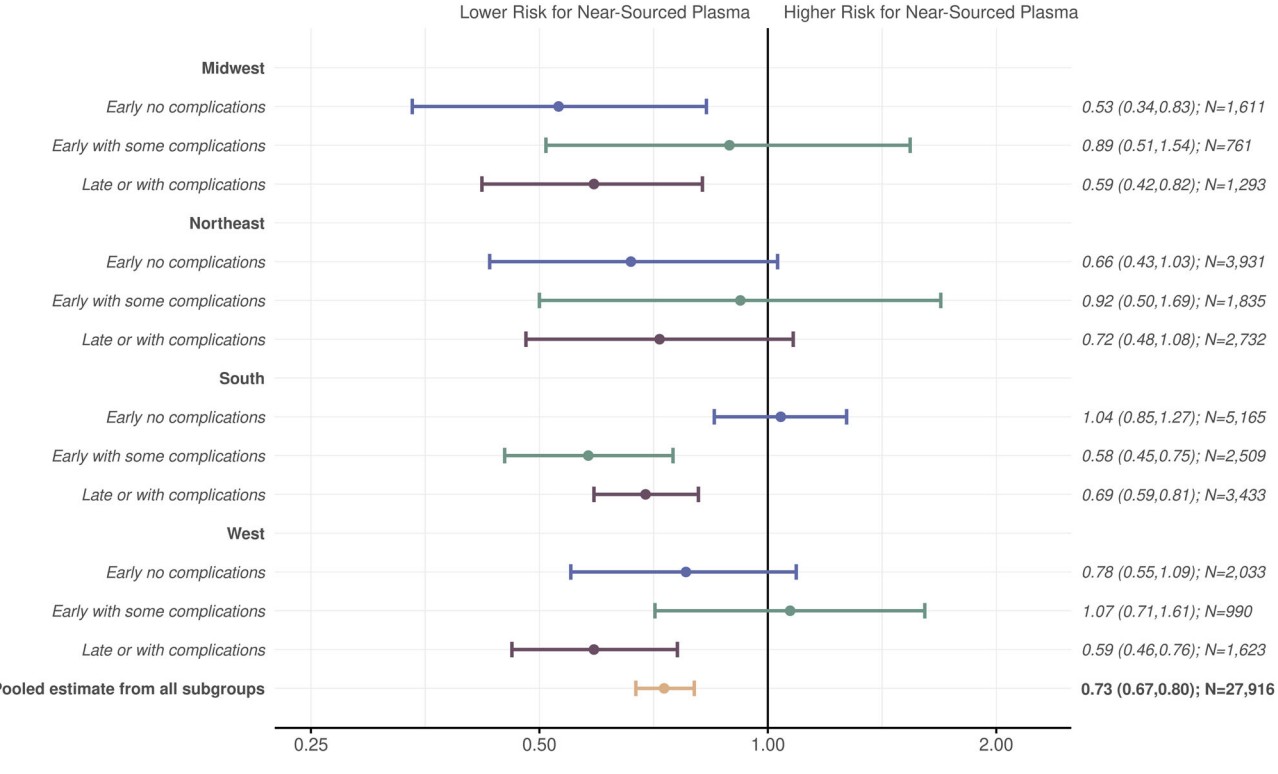

**Fig. 2 Relative risk of death within 30 days after receiving convalescent plasma transfusion from near-sourced plasma vs. distantly sourced plasma.**
This forest plot shows the relative risk of death associated with receiving near-sourced convalescent plasma vs. distantly sourced (≤150 miles vs. >150 miles). The subgroups are the 12 mutually exclusive categories of donor region and patient disease severity. The pooled estimate captures the combined effect across subgroups. Patient disease severity (denoted by color) was defined as follows: Early treatment captures either days to transfusion ≤3 and/or symptom onset to infusion was <7 days, No complications captures no observed risk factors for severe COVID-19 (e.g., respiratory failure; see Supplemental Table 1), Some complications captures 1–2 severe risk factors, and With complications captures 3+ severe risk factors. The pooled estimate from all the subgroups is based on the Cochran–Mantel–Haenszel estimator. Relative risk point estimates are represented by circles and I bars indicate 95% confidence intervals.

mediate protection by inhibiting viral entry to host tissues. Hence, mAb preparations recognize only a few epitopes targeting the virus with high affinity. In contrast, COVID-19 convalescent plasma is composed of multiple immunoglobulins that bind to varying epitopes on the virion with less activity against a single epitope but more activity against many epitopes. Consequently, mAbs have high affinity per protein content at the price of narrow specificity, while COVID-19 convalescent plasma has lower affinity per protein content but a larger antigenic target range. mAb therapies remain protective unless variants emerge that shift their ability to neutralize the virus, as has occurred with several SARS-CoV-2 variants[4,5], leading to their withdrawal from use in several states[18]. In contrast, COVID-19 convalescent plasma is more resilient to single amino acid changes, while its neutralizing antibody efficacy is likely to reflect the overall antigenic composition of the viral population targeted. Thus, mAbs are more likely to be affected by regional differences caused by antigenic shifts in viral strains that affect their respective epitopes than convalescent antibody. Our results demonstrate that convalescent plasma improves mortality in COVID-19 patients if given from local sources and suggests that plasma therapy may continue to provide important therapeutic benefit to combat emerging SARS-CoV-2 strains, particularly if plasma is locally distributed.

## Methods
**Cohort identification**. We analyzed data from the EAP to convalescent plasma for COVID-19[19,20]. The study was approved by the Mayo Clinic Institutional Review

Board (IND 19832 Sponsor: Dr. Michael J. Joyner, MD; ClinicalTrials.gov number: NCT04338360). Written informed consent was obtained from the patients, from legally authorized representatives of the patients, or by means of an emergency consent process if necessary. Patient data were collected using REDCap version 10.6.1 1. Plasma donor neutralizing antibody data were stored in an on-premises DB 2 SQL database.

Regression models and Cochran–Mantel–Haenszel techniques were used to estimate the adjusted relative risk of death within 30 days after transfusion between near-sourced and distantly sourced convalescent plasma. Hospitalized patients with COVID-19 between the ages of 18 and 65 years who were transfused with one or two units of convalescent plasma from a single plasma donor between June 1, 2020 and August 31, 2020 were included in this analysis. Mechanically ventilated patients were excluded because current evidence suggests that convalescent plasma is not effective in this subpopulation[20]. Given that age is a pronounced risk factor for mortality, patients aged >65 years were excluded to further explore the impacts of other potential risk factors[21]. The study period was defined to assess the second wave of enrollment in the EAP cohort to ensure that efficient supply of plasma had been established.

**Statistical analysis**. The allocation of near-sourced vs. distantly sourced plasma was not randomized, so the overall analysis plan was designed to minimize the effects of confounding using multiple complementary statistical approaches. To provide an objective means to identify meaningful differences in demographic, disease, and treatment characteristics between plasma sources, we used standardized mean differences with a cutoff of 10% or 0.10[22] A GBM was used to identify important predictors of 30-day mortality, which included patient characteristics, indicators of disease severity, treatment methods, and distance between COVID-19 patients and plasma donors. The final GBM model was selected by implementing a random grid search tuning across tree depth, number of trees, and learning rate with a stopping criterion of 200 models, and models were ranked based on area under the curve coming from fivefold cross-validation. We investigated multicollinearity among predictors using generalized variance inflation factors from a logistic regression. Results of the GBM were then used to design a series of relative

risk regression models including an unadjusted analysis, adjusted models, and a weighted model with propensity scores from a GBM[23] matching on demographic, disease, and treatment characteristics of importance and using the average treatment effect on the treated estimand[24]. We also examined the relative risk of death at 30 days among subgroups of patient risk factors, time to treatment from COVID-19 diagnosis or symptom onset, and donor region for patients receiving near-sourced plasma vs. distantly sourced plasma. Near-sourced plasma was defined as plasma collected within 150 miles of the transfused patient. This distance of 150 miles was selected as it was considered a reasonable commute time between and within communities and approximated a local area for a given treatment facility. All other plasma was considered distantly sourced.

Data were used as reported in the case report forms, and missing data were not imputed. Analyses were performed with the use of R software[25]. Point estimates for crude mortality were calculated using rates, and 95% CIs were estimated using binomial proportions via the Wilson method. Reported $P$ values are two-sided with $\alpha = 0.05$.

**Reporting summary**. Further information on research design is available in the Nature Research Reporting Summary linked to this article.

## Data availability

Study data cannot be shared publicly because of Institutional Review Board restrictions. Individual participant data underlying the results reported in this publication, along with a data dictionary, may be made available to approved investigators for secondary analyses following the completion of the objectives of the United States Expanded Access Program to COVID-19 convalescent plasma. Limited and de-identified data sets will be deposited into a research data repository and may be shared with investigators under controlled access procedures as approved by the Mayo Clinic Institutional Review Board. A scientific committee will review requests for the conduct of protocols approved or determined to be exempt by an Institutional Review Board. Requestors may be required to sign a data use agreement. Data sharing must be compliant with all applicable Mayo Clinic policies. Interested parties may contact uscovidplasma@mayo.edu.

## Code availability

A data dictionary and custom analysis code may be made available to approved investigators following the completion of the objectives of the United States Expanded Access Program to COVID-19 convalescent plasma. R packages and versions can be provided upon request. Custom code sharing must be compliant with all applicable Mayo Clinic policies. Interested parties may contact uscovidplasma@mayo.edu.

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

## Acknowledgements

This project has been funded in part with Federal funds from the Department of Health and Human Services, Office of the Assistant Secretary for Preparedness and Response, and Biomedical Advanced Research and Development Authority under Contract No. 75A50120C00096. Additionally, this study was supported in part by National Center for Advancing Translational Sciences (NCATS) grant UL1TR002377, National Heart, Lung, and Blood Institute (NHLBI) grant 5R35HL139854 (to M.J.J.) and grant 1F32HL154320 (to J.W.S.), Natural Sciences and Engineering Research Council of Canada (NSERC) PDF-532926-2019 (to S.A.K.), National Institute of Diabetes and Digestive and Kidney Diseases (NIDDK) 5T32DK07352 (to C.C.W.), National Institute of Allergy and Infectious Disease (NIAID) grants R21 AI145356, R21 AI152318, and R21 AI154927 (to D.F.), R01 AI152078 9 (to A.C.), National Heart Lung and Blood Institute RO1 HL059842 (to A.C.), National Institute on Aging (NIA) U54AG044170 (to S.E.B.), Schwab Charitable Fund (Eric E Schmidt, Wendy Schmidt donors), United Health Group, National Basketball Association (NBA), Millennium Pharmaceuticals, Octapharma USA, Inc., and the Mayo Clinic.

## Author contributions

Study conception and design: K.L.K., P.W.J., N.S.P., A.C., R.E.C. and M.J.J. Acquisition, analysis, or interpretation of data: K.L.K., P.W.J., N.v.H., J.W.S., M.M.P., S.A.K., C.C.W., A.M.K., K.A.B., M.R.B., M.A.G., S.E.B., J.R.A.S., K.A.S., M.C.P., D.F., Z.A.B., E.R.W., A.J.C., K.F.L., J.G.R., K.J.A., E.R.L., M.N.P.V., J.J.D. and R.J.R. Drafting of the manuscript: K.L.K., P.W.J., N.v.H., J.W.S., D.F., A.C., R.E.C. and M.J.J. Statistical analysis: K.L.K., P.W.J. and R.E.C. Administrative, technical, or material support: A.M.K., J.R.M., E.S.T., M.A.S., J.C.D.S., N.C.V., P.M., C.M.v.B., J.L.W., J.R.S., R.F.R., V.H., P.R.B. and J.E.B. Supervision: R.S.W., A.C. and R.E.C. All authors contributed to revising the manuscript, and all authors approved the final version of the manuscript.

## Competing interests

The authors declare no competing interests.

## Additional information

[1]Department of Quantitative Health Sciences, Mayo Clinic, Scottsdale, AZ, USA. [2]Department of Quantitative Health Sciences, Mayo Clinic, Jacksonville, FL, USA. [3]Department of Anesthesiology, Cooper Medical School of Rowan University, Cooper University Health Care, Camden, NJ, USA. [4]Department of Anesthesiology and Perioperative Medicine, Mayo Clinic, Rochester, MN, USA. [5]Department of Cardiovascular Medicine, Mayo Clinic, Jacksonville, FL, USA. [6]Department of Laboratory Medicine and Pathology, Mayo Clinic, Rochester, MN, USA. [7]Center for Biologics Evaluation and Research, U.S. Food and Drug Administration, Silver Spring, MD, USA. [8]Department of Epidemiology and Biostatistics, College of Human Medicine, Michigan State University, East Lansing, MI, USA. [9]Department of Pediatrics and Human Development, College of Human Medicine, Michigan State University, East Lansing, MI, USA. [10]Department of Cardiovascular Medicine, Mayo Clinic, Rochester, MN, USA. [11]Human Research Protection Program, Mayo Clinic, Rochester, MN, USA. [12]Department of Internal Medicine, Division of Pulmonary and Critical Care Medicine, Mayo Clinic, Rochester, MN, USA. [13]Department of Internal Medicine, Division of Infectious Diseases, Mayo Clinic, Phoenix, AZ, USA. [14]Department of Molecular Microbiology and Immunology, Johns Hopkins Bloomberg School of Public Health, Baltimore, MD, USA. [15]These authors contributed equally: Katie L. Kunze, Patrick W. Johnson. [16]These authors jointly supervised this work: Arturo Casadevall, Rickey E. Carter, Michael J. Joyner. ✉email: joyner.michael@mayo.edu

