## [Peer Review File · Nature Communications]

Reviewers' Comments:

Reviewer #1:

Remarks to the Author:

The manuscript examines 30 day mortality in 27,952 convalescent plasma-treated patients. They excluded patients who were over 65 years old, and mechanically ventilated patients to exclude those with the highest risk of mortality. They also included only patients who had a single transfusion from one donor. The authors examined the time period of June 1st to August 23rd leaving 27,952 out of a total of 94,287 available for analysis. This is an interesting study and using databases such as these is a real strength.

I am also a big fan of machine learning methods and this database is large enough to justify their use, which is another strength.

The hypothesis of local versus centralized convalescent plasma (hereafter just plasma) based on local variation is an interesting one and an important one to test, especially now as we see increased variation worldwide. It has been shown that the evolutionary divergence worldwide of SARS-CoV-2 began to accelerate in November of 2020. Although, due to low sequencing rates in the US prior to that time, early divergence and important mutations could have been missed. However, the time frame in which this study was conducted is before this evolutionary acceleration began, and before many of the concerning mutations in Spike became more prevalent. Indeed the S477N mutation peaked at around 5% prevalence (as measured by number of each particular mutation deposited into GISIID) worldwide) by September of 2020, E484K 0.01%, N501Y, 0.03% while the D614G mutation was predominant. So while plausible, this hypothesis needs to be carefully tested, as there are no sequencing data provided in the manuscript to test this hypothesis, to ensure that extraneous sources of variability are not confounding the results.

There needs to be more detail in the statistical section. The authors state that they used Gradient Boosted Machines for model selection. There is no mention of hyperparameter tuning. These algorithms are prone to over-fitting and there was no mention of hold-out sample, early-stopping rules, cross-validation etc. It is also known that variable importance can be biased under correlation in tree-based learners and these issues should be addressed.

Age and clinical condition (ICU prior to infusion and Respiratory failure) condition are, by far, the most significant predictors according to the GBM. It is also not surprising that they would be related to mortality. These variables are then followed by US region, which is primarily dominated by the West and South, which accounts for the vast majority of observations. Distance from donor appears to be a continuous variable in the GBM and it seems that 150 miles was chosen as the breakpoint based on the partial dependence plots and as it seems to be a reasonable commute distance. In the regression variable models, I assume this variable was dichotomized as it says local, this should be made clear.

There is also class imbalance and imbalance in the regions as well, with the South and West contributing more patients than the other regions. Is this also true with deaths? It appears that patients in the South were more likely to be treated with non-local plasma versus the North East and the Mid-west. Further it seems that in the South and the West demand for convalescent plasma was outstripping supply. This is most likely due to the number of cases those regions were experiencing during the chosen time frame compared to the other regions. This was not true in the other two regions and yet some of the patients were treated with the non-local plasma. Are some centers more likely to use a central repository versus local? That is, is there a center effect that needs to be controlled for in the analysis? Is there a time effect? That is, were patients treated in the peak (seems to be around early July) more likely to die versus other times and were those patients more likely to be given non-local plasma because of a shortage of supply?

A table of the deaths by plasma source, time regions and demographics would help readers

understand the data and the results.

Reviewer #2:

Remarks to the Author:

This paper used data from US registry to investigate the hypothesis that convalescent plasma has a higher efficacy when the donor and treated patient are in close geographic proximity. The hypothesis is useful for providing practical guidance on treating COVID-19 patients. Through statistical analysis of identifying confounding factors and performing unadjusted and adjusted comparisons among patients receiving near-sourced plasma vs patients receiving distantly-sourced plasma, the authors identified lower risk of within 30 days mortality in the near-sourced plasma group. The authors provided justifications from the perspective of the biology of SARS-CoV-2 and the immunology of COVID-19. Regarding the statistical analysis part, I have the following comments.

1. My major concern is how the confounding variables were adjusted in the statistical analysis. The authors mentioned that GBM was used to identify important predictors of 30-day mortality and based on Figure S2, it is unclear what variables are eventually included as confounding factors. Given the current sample size and the number of candidate variables in Figure S2, I wonder if performing GBM to screen variables is indeed necessary.
2. After dichotomizing distance to donor into the "near-sourced plasma" and the "distantly-sourced plasma" groups, a better way to control for the confounding effects is to do a propensity score matching or stratification. It is important to rigorously quantify the differences of confounding variables between the two groups. The author only argued that "Baseline characteristics were similar across distance cohorts except for geographic region and race/ethnicity".
3. Please provide p-values in Table S1 comparing each variable between the "near-sourced plasma" and the "distantly-sourced plasma" groups.
4. It is unclear what proportion of data are missing and if "missing data were not imputed", how were they handled in this study?
5. The main paper should be divided into "Introduction, Result and Discussion" to better assist readers to locate information they are interested in.

RESPONSE TO REFEREES

We are grateful to all reviewers for their efforts to improve our manuscript. We are appreciative of the many insightful comments and helpful suggestions to improve our analyses specifically and the manuscript overall. With this in mind, **four main issues** were identified and are addressed both here generally and below in response to specific reviewer comments.

First, the reviewers have suggested using a propensity score matching model in order to minimize the influence of any potential confounding effects. We appreciate this suggestion and have included an additional relative risk regression model in Table S3 (Model 4). The model we used was a gradient boosted model to estimate propensity scores and weighting. Results showed a statistically significant difference in the relative risk of death within 30 days of transfusion for near-sourced vs. distantly-sourced plasma. Additionally, we performed a Cochran-Mantle-Haenszel stratified analysis based on region and propensity score. These results were consistent with the regression analysis and have been included below. Finally, we have reported standardized differences in Table S1 to allow for more-straightforward comparisons between groups receiving near- vs. distantly-source plasma.

Second, the reviewers have questioned the fit of the gradient boosting machine used herein. As hyperparameters were chosen based on previous work by the authors and not tuned for this specific analysis, we agreed that further investigation was warranted. In our attempts to answer their questions completely and report measurements of model fit, we discovered that our previously reported model was not the model which best fit the data. As such, we have elected to include this updated, better-fitting model in our reporting. The results of this GBM do not differ significantly in which variables were of greatest importance and the variable importance plot in Figure S2a shows donor distance as the fourth most important variable (up from 5th most important variable in our previous-reported model). Our relative risk regression models were not impacted by this change. We do, however, report a much larger range of estimated mortality at 30 days post-infusion observed across distance in the partial dependence plot included in an updated Figure S2b.

Third, a comparison of the mortality rates between patients receiving near- and distantly-sourced plasma across variables of interest was warranted and has now been included in Table S2.

Fourth, we have addressed concerns raised by both the editor and Reviewer 1 regarding our choice of cutoff for classification of near-sourced vs. distantly-sourced plasma in detail in Response 5 to Reviewer 1. As stated, we determined this threshold a priori based on an estimate of a reasonable commute time and/or the transfer of plasma between collection and infusion sites. This decision was supported by the partial dependence plot in the GBM and a sensitivity analysis conducted at a distance cutoff of 250 miles.

We hope these broad responses above along with our detailed point-by-point responses below are useful in the review of our manuscript. We feel that these additions and changes have further emphasized the importance of the distance between plasma donor and recipient for outcomes of greatest interest such as rates of mortality. We are grateful to the reviewers for their comments which have greatly enhanced the quality of this work and the potential impact of our findings. Thank you for your careful consideration and valuable feedback.

REVIEWER 1 COMMENTS

The manuscript examines 30 day mortality in 27,952 convalescent plasma-treated patients. They excluded patients who were over 65 years old, and mechanically ventilated patients to exclude those with the highest risk of mortality. They also included only patients who had a single transfusion from one donor. The authors examined the time period of June 1st to August 23rd leaving 27,952 out of a total of 94,287 available for analysis. This is an interesting study and using databases such as these is a real strength.

Response 1: We thank Reviewer 1 for their careful review of the manuscript. We have provided a detailed response to the questions that follow and have indicated relevant changes in the manuscript where applicable.

I am also a big fan of machine learning methods and this database is large enough to justify their use, which is another strength.

Response 2: Thank you. We were also pleased to have a dataset large enough to implement machine learning methods.

The hypothesis of local versus centralized convalescent plasma (hereafter just plasma) based on local variation is an interesting one and an important one to test, especially now as we see increased variation worldwide. It has been shown that the evolutionary divergence worldwide of SARS-CoV-2 began to accelerate in November of 2020. Although, due to low sequencing rates in the US prior to that time, early divergence and important mutations could have been missed. However, the time frame in which this study was conducted is before this evolutionary acceleration began, and before many of the concerning mutations in Spike became more prevalent. Indeed the S477N mutation peaked at around 5% prevalence (as measured by number of each particular mutation deposited into GISAID) worldwide) by September of 2020, E484K 0.01%, N501Y, 0.03% while the D614G mutation was predominant. So while plausible, this hypothesis needs to be carefully tested, as there are no sequencing data provided in the manuscript to test this hypothesis, to ensure that extraneous sources of variability are not confounding the results.

Response 3: We appreciate your comments regarding the timing of variants of interest. We do not have any genetic classifications of the SARS-CoV-2 in plasma donors or plasma receiving patients in the current study. We would like to note that research by Rochman et al.¹ cited in our manuscripts shows evidence of known variants circulating in the U.S. as early as July 2020 and posits that meaningful variants were likely circulating prior to the time of our cohort. We have adjusted the language in lines 71-72 to reflect that the circulation of these variants is not confirmed.

1. Rochman, N.D., et al. Ongoing Global and Regional Adaptive Evolution of SARS-CoV-2. bioRxiv (2021).

There needs to be more detail in the statistical section. The authors state that they used Gradient Boosted Machines for model selection. There is no mention of hyperparameter tuning. These algorithms are prone to over-fitting and there was no mention of hold-out sample, early-stopping rules, cross-validation etc. It is also known that variable importance can be biased under correlation in tree-based learners and these issues should be addressed.

Response 4: Our original approach was to use the same model architecture designed in previous publications². After considering the reviewers' responses, we agree a more contrived approach could have been taken. To address these concerns, we implemented a random grid search tuning across tree depth, number of trees, and learning rate with a stopping criterion of 200 models. Models were ranked based on area under the curve (AUC) coming from 5-fold cross-validation and the top model was then compared to the previously reported GBM. Performance was significantly better ($p < 0.001$ for DeLong's test for correlated AUCs), so we have opted to use this model in the manuscript. We have also updated the manuscript to include these details in the statistical methods section. We thank the reviewer for this suggestion and the opportunity to make this improvement.

In regards to the variable importance plot, we agree with the ordering and importance could be influenced by multicollinearity amongst the predictors. As a sensitivity analysis, we fit a conditional random forest model which provides a more objective ranking of correlated predictors as shown in Strobl (2008)³. Due to large amount of missing data for medications and Ortho IgG that arose from changes in EAP database over time and availability of sample, respectively, these variables were removed in the conditional random forest model set of candidate variables. The top 5 variables in this analysis were also in the top 5 of the GBM (however, the fourth and fifth ranked variables were transposed between the two models (US region and donor distance). Additionally, variance inflation factors for all predictors were calculated. Baseline condition, which had a near-zero variable importance, was the only predictor with a value over 2. Our analytical approach used 3 models to limit the impact of multicollinearity, but we did not mention in the text for brevity.

These changes are reflected in lines 247-255 of the updated manuscript.

2. Joyner, M.J., et al. Convalescent Plasma Antibody Levels and the Risk of Death from Covid-19. *N Engl J Med* (2021).
3. Strobl, C., Boulesteix, A.-L., Kneib, T., Augustin, T. & Zeileis, A. Conditional variable importance for random forests. *BMC Bioinformatics* 9, 307 (2008).

Age and clinical condition (ICU prior to infusion and Respiratory failure) condition are, by far, the most significant predictors according to the GBM. It is also not surprising that they would be related to mortality. These variables are then followed by US region, which is primarily dominated by the West and South, which accounts for the vast majority of observations. Distance from donor appears to be a continuous variable in the GBM and it seems that 150 miles was chosen as the breakpoint based on the partial dependence plots and as it seems to be a reasonable commute distance. In the regression variable models, I assume this variable was dichotomized as it says local, this should be made clear.

Response 5: We added footnotes to Supplementary Table 3 (previously Table S2, see responses below for explanation) to clarify the distance variable. In regard to the 150 miles limit, we also discussed this as part of the development of the study prior to any modeling and chose a threshold that would approximate a local area and/or reasonable travel distance for disbursing plasma or for an individual to travel in order to seek care during the pandemic. In the GBM, we observed a leveling off at 250 miles in the partial dependence plot but chose to continue using our a priori threshold of 150 miles. As a sensitivity analysis, we reran our relative risk regression models with the cutoff of 250 miles and observed similar estimates, significance levels, and interpretation of results. We have chosen to use our previously-determined cutoff of 150 miles in the manuscript.

There is also class imbalance and imbalance in the regions as well, with the South and West contributing more patients than the other regions. Is this also true with deaths? It appears that patients in the South were more likely to be treated with non-local plasma versus the North East and the Mid-west. Further it seems that in the South and the West demand for convalescent plasma was outstripping supply. This is most likely due to the number of cases those regions were experiencing during the chosen time frame compared to the other regions This was not true in the other two regions and yet some of the patients were treated with the non-local plasma. Are some centers more likely to use a central repository versus local?

Response 6: The EAP collected limited information on the participating sites. We suspect that indeed some centers are more likely to use a central repository. This type of analysis is not feasible for our data because substantial subsetting would be needed to examine cases, controls, and deaths, which would create issues with small cell sizes with 91% of sites enrolling fewer than 50 patients. See Response Figure 1 for additional detail of number of patients per site.

Response Figure 1

That is, is there a center effect that needs to be controlled for in the analysis? Is there a time effect? That is, were patients treated in the peak (seems to be around early July) more likely to die versus other times and were those patients more likely to be given non-local plasma because of a shortage of supply?

Response 7: We did observe differing mortality rates across the implementation of the EAP. Treatment month was accounted for in our GBM analysis. This GBM was designed with a maximum depth of 3 to allow for this type of interaction to be observed. Response Figure 2 shows PDP plots for distance stratified region and transfusion month. A consistent trend was observed across all stratifications indicating the impact of near-sourced vs. distantly-sourced plasma does not vary across time or region. In addition, the original CMH analysis presented in the paper was also stratified by region and other key variables of interest. The consistency of estimates and trends across these modeling approaches was interpreted favorably as we reported.

Response Figure 2

A table of the deaths by plasma source, time regions and demographics would help readers understand the data and the results.

Response 8: We have created the table of mortality within 30 days of plasma transfusion as suggested including key demographic variables, treatment month, and region. Please see what is now labeled as Supplemental Table 2. The table previously labeled as Table S2 is now labeled as Table S3 and all references have been updated in the body of the manuscript.

REVIEWER 2 COMMENTS

This paper used data from US registry to investigate the hypothesis that convalescent plasma has a higher efficacy when the donor and treated patient are in close geographic proximity. The hypothesis is useful for providing practical guidance on treating COVID-19 patients. Through statistical analysis of identifying confounding factors and performing unadjusted and adjusted comparisons among patients receiving near-sourced plasma vs patients receiving distantly-sourced plasma, the authors identified lower risk of within 30 days mortality in the near-sourced plasma group. The authors provided justifications from the perspective of the biology of SARS-CoV-2 and the immunology of COVID-19. Regarding the statistical analysis part, I have the following comments.

Response 1: We thank Reviewer 2 for their careful consideration of our work. We have provided a detailed response to the questions that follow and have indicated relevant changes in the manuscript where applicable.

My major concern is how the confounding variables were adjusted in the statistical analysis. The authors mentioned that GBM was used to identify important predictors of 30-day mortality and based on Figure S2, it is unclear what variables are eventually included as confounding factors. Given the current sample size and the number of candidate variables in Figure S2, I wonder if performing GBM to screen variables is indeed necessary.

Response 2: All potential confounding factors for the regression analyses were included in Table S2. The Base Model included no confounders, Model 2 included demographic data, disease status, time to transfusion and region. Model 3 included these confounders plus additional disease and treatment predictors. Data were not collected for medications throughout the entire EAP, which limited their usability in all models and significantly reduced the available sample size in Model 3.

The question of whether or not a GBM is necessary is a fair question. We elected to use it in several publications to allow for observations with missing data, which occurred by design with changing data collection instruments, as well as to provide visuals of the drivers of mortality that we could review. We recognize that there are other approaches that could be considered, but we have found this one to be successful.

After dichotomizing distance to donor into the “near-sourced plasma” and the “distantly-sourced plasma” groups, a better way to control for the confounding effects is to do a propensity score matching or stratification. It is important to rigorously quantifying the differences of confounding variables between the two groups. The author only argued that “Baseline characteristics were similar across distance cohorts except for geographic region and race/ethnicity”.

Response 3: Standardized differences are now included in Table S1 to highlight the differences in baseline characteristics across the two distance categories. We used 10% difference as rule of thumb for interpreting the magnitude of the effect⁴. Additional detail has been added to the Results section on lines 94-98 and the Online Methods section on lines 244-245. We have also made a small typographical edit to Table S1 to more clearly denote what is being captured in the Gender Other category.

We have also calculated propensity scores based on a GBM⁵. In one analysis, we divided the propensity scores into subcategories and ran a Cochran-Mantle-Haenszel analysis to look for consistency of the treatment effect over the range of scores. Additionally, we used the weights from the GBM in a relative risk regression. The Online Methods has been updated on lines 255-258 to include these additional details.

The results of the relative risk regression based on this model are now included as Model 4 in Supplemental Table 3. We have provided the additional CMH analysis in Response Figure 3 for the reviewers but do not feel it should be included in the main body of the paper as the results are consistent with our original analysis and the added Model 4.

4. Austin, P.C. Balance diagnostics for comparing the distribution of baseline covariates between treatment groups in propensity-score matched samples. *Stat Med* 28, 3083-3107 (2009).
5. McCaffrey, D.F., Ridgeway, G. & Morral, A.R. Propensity score estimation with boosted regression for evaluating causal effects in observational studies. *Psychol Methods* 9, 403-425 (2004).

Response Figure 3

Please provide p-values in Table S1 comparing each variable between the “near-sourced plasma” and the “distantly-sourced plasma” groups.

Response 4: The authors chose not to display p-values as the interpretability of p-values decreases with large sample sizes; however, the added standardized differences were included in Table S1 to help the reader quantify the differences (on lines noted in Response 3 to Reviewer 2).

It is unclear what proportion of data are missing and if “missing data were not imputed”, how were they handled in this study?

Response 5: To capture rates of missing data, the numerator and denominator for each cell were included in Table S1. For COVID-19 risk factors, only patients with severe/life threatening COVID-19 were asked to provide this information. For medications, only those who reported one of these medications are included in the analysis. Other missing data included 26 patients who had a missing weight status and 53 patients who were treated in outlying U.S. territories which are not assigned a U.S. Census Region (these were included in the distance analysis but cannot be reported as being within a specific region). In the Statistical Analysis section, it is mentioned data were not imputed; however, the methods we selected such as the GBMs are robust to missing data and therefore no imputations or cohort subsetting was needed. In the regression analyses and CMH, only patients with complete data were used for analysis. The sample size for each model is reported with the results of the model in Supplemental Table 3.

The main paper should be divided to into “Introduction, Result and Discussion” to better assist readers to locate information they are interested in.

Response 6: Thank you for the suggestion, we agree this improves readability, and the headings have been added. These titles were left out by error and now reflect the guidelines put forth by Nature Communications.

ADDITIONAL MODIFICATIONS

Additional change 1: Small typographic changes were made to Table S3 to align variable labels across tables such that “ICU Prior to Infusion” now reads “ICU care before infusion” and “Donor Distance (Local)” now reads “Donor Distance (Near-sourced)”. A more-detailed footnote is now included to provide additional information on model frameworks used and the dichotomization of donor distance.

Additional change 2: Supplemental Figure 1 was updated to correct an error on the reported date for “Last Transfusion Included”. It now reads August 31, 2020 which matched descriptions in text. Additional clarification was added to the figure description to match what was reported in text.

Additional change 3: The x-axis of the variable importance plot (Figure S2a) has been rescaled to be between 0 and 1. The y-axis of the partial dependence plot (Figure S2b) has been rescaled to capture the range of estimated mortality rates expressed as percentages for the new GBM.

Additional change 4: For consistency, references to “near-sourced” and “distantly-sourced” were correct throughout the document where other labels had previously been used in error.

Additional change 5: A handful of additional typographical and grammatical changes have been made throughout.

Additional change 6: Data and code sharing statements and ClinicalTrials.gov number have been included as required.

Reviewers' Comments:

Reviewer #1:

Remarks to the Author:

The authors are responsive to review. I recommend accepting the paper.

Reviewer #2:

Remarks to the Author:

I am satisfied with the revision made on the manuscript.